# Survey of Reinforcement-Learning-Based MAC Protocols for Wireless Ad Hoc Networks with a MAC Reference Model

**DOI:** 10.3390/e25010101

**Published:** 2023-01-03

**Authors:** Zhichao Zheng, Shengming Jiang, Ruoyu Feng, Lige Ge, Chongchong Gu

**Affiliations:** College of Information Engineering, Shanghai Maritime University, Shanghai 201306, China

**Keywords:** MAC reference model, wireless ad hoc network, medium access control protocols, reinforcement learning

## Abstract

In this paper, we conduct a survey of the literature about reinforcement learning (RL)-based medium access control (MAC) protocols. As the scale of the wireless ad hoc network (WANET) increases, traditional MAC solutions are becoming obsolete. Dynamic topology, resource allocation, interference management, limited bandwidth and energy constraint are crucial problems needing resolution for designing modern WANET architectures. In order for future MAC protocols to overcome the current limitations in frequently changing WANETs, more intelligence need to be deployed to maintain efficient communications. After introducing some classic RL schemes, we investigate the existing state-of-the-art MAC protocols and related solutions for WANETs according to the MAC reference model and discuss how each proposed protocol works and the challenging issues on the related MAC model components. Finally, this paper discusses future research directions on how RL can be used to enable MAC protocols for high performance.

## 1. Introduction

The wireless ad hoc network (WANET), as a wireless communication network with frequently changing topology [1], has attracted extensive attention among scholars. A large number of mobile nodes (MNs) with functions such as information perception, data processing and wireless communication constitute a WANET environment. Therefore, the communication medium is often shared by multiple nodes at the same time and controlled with medium access control (MAC) protocols. A WANET is a decentralized type of wireless network not relying on a preexisting infrastructure [2,3]. Maintaining the reliability and security of wireless networks (WNs) is especially difficult, compared to the infrastructured device because of the mobility of nodes. Moreover, due to the hidden node problem, signal collision occurs frequently when two nodes do not know the existence of each other and need to access the shared media at the same time. The wireless network topology is highly dynamic because wireless links are very fragile due to path loss, poor interference immunity and the mobility of MNs. Moreover, whenever congestion occurs at intermediate nodes, packet loss and delays accumulate, which makes it impossible to obtain satisfactory network performance. It is a pressing demand for the development of efficient intelligent schemes to improve the performance of WANETs.

In highly dynamic WNs, MAC protocols are also required to be designed to adapt to changing conditions [4,5]. In the last several years, machine learning (ML) has been widely applied in solving different application-oriented problems. It allows the agent to learn different rules through statistics and different data sets by applying artificial intelligence (AI) to handle the design and development of algorithms and models [6,7]. ML is suitable for solving network problems due to the following two reasons: its capability of interacting with the environment from input information and its capability of making control decisions, i.e., wireless devices collect large quantities of observations, then feed them into ML algorithms to help make better decisions. For example, a series of techniques are used to design and optimize nodes in dynamic scenes to adapt to constantly changing environments. Some solutions are dedicated to rescue operations [8], forest fire monitoring and prevention [9]. Similarly, there are some works for underwater acoustic networks (UANs) [10,11,12], the Internet of things (IoT) [13,14,15,16], unmanned aerial vehicle (UAV) formations [17,18,19,20,21], vehicular ad hoc networks [22], flying ad hoc networks (FANET) [23,24,25], heterogeneous networks (HetNet) [26,27,28,29] and next-generation wireless communication [30,31,32]. In addition, there are some works aimed at security [33], robustness [34], energy saving [35,36], adaptability [37,38,39,40,41,42,42] and stability [43,44,45,46,47,48,49,50,51]. As recent WNs become more complex, more demands are placed on learning systems [52]. In such dynamic networks, intelligent MAC schemes should be designed to adapt to changing topologies. Instead of switching between different solutions, reinforcement learning (RL) is used to continuously evaluate channel conditions by inquiring and responding to a channel state in order to maintain a conflict-free schedule.

Considering the important role of intelligent model in addressing the above challenges, this paper presents an overview of intelligent applications at the MAC layer with a comprehensive understanding of ML-based MAC protocols under a MAC reference model. We focus on RL [53] and deep reinforcement learning (DRL) [54] approaches in the context of WANETs. The main contributions of this paper are summarized as follows:We describe how to use RL to classify observations into positive/negative types, and how to integrate it with different MAC components, to make a better decision.We review different RL algorithms used in various MAC components.We summarize open research issues and future directions in this field.

As shown in Figure 1, the rest of the paper is organized as follows. Section 2 discusses existing survey articles in this area, with some classic RL schemes introduced in Section 3. We classify the reviewed RL-based MAC protocols according to the MAC reference model [55] and summarize them in Section 4. Then, we present research challenges to improve the MAC design and discuss further research issues in Section 5. Finally, we conclude this study in Section 6.

## 2. Review of Related Survey Articles

Through a comprehensive survey of the intersection of the two fields, Ref. [56] bridges the gap between deep learning (DL) and mobile wireless network research by introducing DL methods and their potential applications in mobile networks. It also reviews various technologies that can be used to deploy DL on WNs, such as data analysis, network control, signal processing, etc. In addition, it also provides a comprehensive list of papers related to mobile/WNs and DLs and classifies them according to different application fields.

Ref. [57] introduces the latest application of ML in improving the performance of MAC throughput, energy efficiency and delay in wireless sensor networks. It comprehensively reviews the application of ML technology and supports the implementation of real-time-based Internet of things (IoT) applications by optimizing the performance of MAC protocols. A similar work was conducted in [58]. In order to reduce the energy consumption of IoT devices and achieve a higher throughput, this paper summarized the RL-based time-division medium access (TDMA) MAC protocol and compares them from several unified features.

Ref. [59] surveys ML models, radio and MAC analysis and network prediction approaches in WNs, considering all layers of the protocol stack: physical (PHY), MAC and network. A comprehensive review is conducted on the use of ML-based schemes to optimize wireless communication parameter settings to achieve an improved network quality of service (QoS) and quality of experience (QoE). Finally, research challenges and open issues are discussed. Ref. [32] also conducts an in-depth survey on recent network optimization AI methods to ensure end-to-end QoS and QoE in 6G. It introduces the ML-based works of end-to-end transport flows from network access, congestion control and adaptive flow control. Some outstanding problems and potential research directions are discussed. Refs. [30,31] also survey multiple access schemes for next-generation wireless systems and give a detailed comparisons of 1G to 6G.

Ref. [60] surveys the latest works of ML applications in WNs by classifying them into three types according to different protocol stacks: resource control in the MAC layer, routing and mobility management in the network layer and localization in the application layer. In addition, several conditions of applying ML to wireless communication are analyzed, and the performance comparison between the traditional methods and ML-based methods is summarized. Finally, challenges and unsolved problems are discussed.

The reconfigurable intelligent surface (RIS) technology is used to reconstruct the wireless propagation environment by integrating the characteristics of large arrays with inexpensive antennas based on metamaterials. However, the MAC of RIS channels supporting multiuser access is still in the preliminary stage of research. Ref. [61] elaborates the architecture of the integration of a RIS and a MAC protocol in a multiuser communication system, introducing four typical RIS-assisted multiuser scenarios. Finally, some challenges, prospects and potential applications of RIS technology related to MAC design are discussed.

Ref. [62] outlines various ML-based methods in wireless sensor networks (WSNs) for smart city applications. The ML methods in WSN-IoT was investigated in detail, including supervised and unsupervised learning algorithms and RL algorithms.

ML is expected to provide solutions to various problems of UAVs used for communication. All relevant research works have been investigated in detail in [63], which have been used in UAV-based communications from the aspects of channel control, resource allocation, positioning and security. Table 1 compares our survey with the discussed references.

## 3. RL Algorithms

With ML, computers learn to perform tasks by learning a set of training examples training, then uses the new set of data to perform the same task and give the results. In this section, we give a brief introduction on the ML framework, and the differences and relationship between RL and DRL are discussed.

### 3.1. Machine Learning Framework

One of the most commonly used ML algorithms is a neural network (NN), which loosely models neurons in biological brains. A typical NN system consists of two parts: neuron and connection. Each connection has a weight, which determines how strongly a node at one end affects another [64].

### 3.2. Reinforcement Learning

Through continuous learning and self-correction, humans spontaneously interact with their surroundings to accomplish tasks, such as object recognition, adjustment and selection [65]. Inspired by human behavior, RL was proposed. According to the different operating mechanisms, RL can be divided into the following two types: model-based RL and model-free RL [65]. The model-based RL algorithm allows an agent to learn a model to describe how the environment works from its observation and then uses this model to make action decisions. When the model is very suitable for the environment, the agent can directly find the optimal strategy through learning algorithms. However, some tasks cannot be modeled. The model-free RL scheme is an effective solution. In this scheme, the agent does not try to model the environment, but updates its knowledge based on the value iteration method to estimate the future return, so as to understand the value of actions taken in a given state. According to the strategy updating method, the model-free RL algorithm can be divided into round-update algorithms and step-update algorithms. Policy optimization and Q-learning are widely used in step-update algorithms. An RL system consists of the following three parts: an agent which interacts with the environment, an environment from which can be extracted some features and an AI algorithm which processes data and makes decisions. The AI algorithm makes decisions through an agent to influence the state and makes decisions again through feedback given by the environment, as shown in Figure 2a.

*Q*-learning [66] is a widely used RL algorithm. A *Q*-table Q(s,a) is the core and the expectation that obtain a reward by taking some action *a* in the state *s* over a period of time. The corresponding reward *r* is obtained from the feedback of the environment according to agent action *a*. Therefore, the key idea of *Q*-learning is to construct a table composed of states and corresponding actions to store the *Q*-value, and then selects one action that can obtain the maximum reward. Through continuous trial and error, an RL system aims to find a law from the mistakes and learn to achieve the goal without any idea at first. In addition, the actor–critic algorithm combines two kinds of reinforcement learning algorithms based on a value (such as Q learning) and policy (such as policy gradients).

A learning automaton (LA) is an RL algorithm that operates in a probability space and learns the optimal value by continuously interacting with an unknown environment. LA algorithms do not need to know their usage scenarios or any knowledge about the target to be optimized is a major advantage [67]. In recent years, some works [21,67,68] have attempted to apply LA to solve various practical problems, e.g., optimizing congestion-aware scheme and configuring the LA in the routing node.

### 3.3. From RL to DRL

As modern WNs become more and more complex, learning systems are required to have better performance, such as more powerful computing power, faster learning, better robustness and more extensible input and output mechanisms [52]. Therefore, DL applications for WNs have aroused widespread interest. Compared to a traditional RL algorithm, DL offers more improvements in WANET applications with more complex and large input data, by using multiple hidden layers between the input and output layers. Due to the increase of the action space and the state space from the input data, the *Q*-table is too huge to calculate. To address the above problems, a deep *Q*-learning (DQN) algorithm combines DL with *Q*-learning and uses the function instead of *Q*-tables to fit the relationship between the action and the state. A DQN algorithm extracts the features of the input state as the input, then calculates the value function through an NN as the output and trains the function parameters until convergence. The schematic structure of a DQN is shown in Figure 2b. The input data are accepted by the input neurons and transmitted to the hidden layer neurons. An agent obtains output data through the output layer after the training phase. The agent takes the result of the algorithm and computes the error by comparing it to the target model [69]. The policy is also updated using backpropagation by the agent. After a certain time, the agent adopts various actions and saves the current status, rewards obtained, next status and actions adopted into the experience replay memory (ERM), which is employed into the DQN framework to avoid potential nonconvergence of the gradient calculation. This makes the algorithm converge faster and more stably, and the agent can make decisions efficiently using the learned model. A taxonomy of RL is shown in Figure 3.

### 3.4. Optimal Design with RL

WNs have complex characteristics in their communication signals, channel quality, node topology, link congestion, etc. [70]. Moreover, the communication performance is significantly affected by MAC protocol targets, such as access delay, fairness, energy efficiency, congestion control, etc. The relationship between each state in the network environment satisfies the Markov property, and the stochastic process can be regarded as a Markov process (MP). When RL schemes are used to make decisions for the protocol, the reward and the action are introduced into the system, thus turning the MP problem into a Markov decision process (MDP). RL schemes have been widely developed to handle complicated wireless network situations with superiority over conventional schemes as listed below.

Interaction with the environment: Due to node mobility, the network topology is constantly changing so that protocols are also required to be dynamically and automatically configured to achieve better network performance. RL can interact with the environment and feed back network changes to the protocol in time.Self-learning and self-correction: The characteristics of self-learning and interaction with the surroundings make systems adapt to the frequent changes of the network in time. Thus, the optimal scheme should be more adaptive, highly scalable and energy-efficient.Model without prior training: WNs are generated at any time and easy to construct; they do not need to be supported by preexisting networks. Compared with other learning schemes using trained models, RL schemes train the model by interacting with the networks in real time to make the trained model more suitable for the current environment.

## 4. Application of Reinforcement Learning for MAC Protocols

Several papers have investigated the applications of RL in MAC protocol design to provide accelerated data processing while maintaining a high throughput, with a minimization of collision and time saving. The review here adopts the MAC reference model proposed in [55], which compartmentalizes a MAC protocol into the following three components: operation cycle (OC), medium access unit (MAU) and MAC mechanism (MM). At the beginning of an OC, some MMs start running, which can be used to compete for the channel, and then, node accesses the medium with certain MAUs for transmission. Various MMs have different running sequences, the exchanged message content, the capacity and number of MAUs available in each OC constituting the MAC protocol process, which vary according to the function and design of different protocols [55]. The architecture diagram of MAC protocol components is illustrated in Figure 4. These components form the general structure of various MAC schemes for easier understanding.

Following this model, how RL schemes are used to improve the performance and effectiveness of these components are discussed below.

### 4.1. Operation Cycle

A MAC operation cycle is a repeated time epoch with either a fixed format or a random interval, which includes general frames, large frames and superframes [55]. One approach of using RL in MAC protocol design is modifying the OC, such as the length of the general frame and the structure of the frame. Modifying the superframe has been used in [46] for the purpose of protocol design and optimization.

The time slot is divided into a normal time slot and an emergency time slot, such as the superframe. Traditional MAC protocols lack efficient transmission rate control and backoff strategies in emergency transmission. This is a challenge for reliable emergency data transmission. To overcome these problems, an enhancement objective *Q*-Learning MAC (eOQ-MAC) protocol for emergency transmission was proposed in [46]. A novel superframe structure was designed for a slotted frame structure in eOQ-MAC, which modified the MAC frame by adding a new type of slot in it and selected a time slot to transmit data through a distributed slot selection algorithm with *Q*-Learning. If an emergency message needed to be sent, it automatically used the emergency time slot. If there was a data collision in the emergency time slot, the node without emergency data left its emergency time slot. The simulations showed that this packet loss rate of modification was lower than CSMA/CA’s 79.6% in emergency transmissions.

### 4.2. Medium Access Unit

A medium access unit is the fundamental unit for nodes to access the shared medium. It can be determined not only by the MAC scheme, e.g., MAC frames, time slots and minislots, but also by the multiplexing solution in the physical layer, e.g., time slots, code and sub-bands [55]. The second approach of using RL in MAC protocol design is modifying a MAC unit or multiplexing unit in the MAU, such as the channel access method [10,45].

In [10], the author proposed a slotted carrier sensing multiple access (CSMA) protocol, using RL to extend the life cycle of nodes in underwater acoustic wireless sensor networks (UA-WSNs), which have energy constraints due to the complex environment, where power replacement and charging for underwater devices are difficult and expensive. Meanwhile, the channels are vulnerable and changing with a poor delay and limited available bandwidth [55]. The underwater acoustic channel was divided into several subchannels. The design of the protocol mainly considered three factors: the transmission power, the number of shared channel neighbors and the selection of subchannels. These factors were considered to design the reward function in the protocol, using *Q*-learning to adjust the time slot length and choose a suitable subchannel to ensure the success of the transfer. It was shown that the proposed scheme was able to extend the lifetime of the network by adjusting the parameters of the algorithm to adapt to the underwater environment and reduce energy as much as possible.

Ref. [45] designed an adaptive MAC scheme for WSNs with the irregular repetition slotted ALOHA (IRSA) protocol. Due to uncertain and changing channel conditions in WSNs, especially the waterfall effect in the traditional IRSA protocol, *Q*-learning was used there to allocate a limited frame size and improve the common channel throughput. It tried to find the optimal policy that optimized expected rewards for all actions by learning the input–output relationship. RL-IRSA was optimized for small frame sizes, and the global throughput for high channel loads was effectively improved.

HetNet is a large-scale network with more access competition. Ref. [28] proposed a novel DRL-based MAC scheme for HetNet, named deep-reinforcement learning multiple access (DLMA). By correctly defining the state space, action space and rewards in a novel multidimensional DRL system, this scheme could maximize the total throughput and ensure the fairness of agents by selecting a specific time slot for transmission. The work in [71] also proposed an RL-based MAC protocol to select the appropriate time slot to access the network for slotted CSMA. By collecting collision conditions among slots, the RL system could effectively reduce collisions and packet delay. The above-discussed MAC protocols are summarized in Table 2.

### 4.3. MAC Mechanism

A MAC mechanism is the competitive action taken by nodes expecting access to the medium, which can be divided into basic mechanisms and complex mechanisms (combination of basic mechanisms) [55]. Some significant basic mechanisms are listed below: free access, slot access, backoff, signaling, carrier sensing, messaging, scheduling, prioritization, etc. The third approach of using RL in MAC protocol design is modifying the MM for collision avoidance. MMs are modified and controlled by RL schemes to access the channel and find approximate solutions for optimization.

#### 4.3.1. Backoff

In [8], the author proposed a fully distributed rate adaptive CSMA/CA protocol, which was an energy-saving multiuser scheduling MAC protocol based on RL for delay-sensitive data. Enabling the system to effectively trade off consumption and delay by using traditional CSMA/CA schemes is challenging. In order to balance consumption and delay, the energy-saving and delay-sensitive multiuser resource allocation problem was formulated as an MDP. Then, an RL algorithm was used to solve the agent’s access task, and an optimal policy for each agent was learned through interaction with the environment. Unlike traditional CSMA/CA, this scheme adaptively changed the congestion windows (CWs) so that a node could increase the rate of transmission if the queue buffer was insufficient. Additionally, the proposed MAC protocol assumed that the backoff counters would be updated to fit its current situation instead of freezing it when multiple nodes grabbed the channel. Compared with the traditional CSMA/CA, this solution enabled nodes to achieve tighter latency limits with an energy consumption of 53%. Moreover, the proposed learning algorithm could be updated multiple times in each time slot to speed up the convergence speed and improve runtime efficiency.

In [72,73,74,75,76], a CW size adjustment was also used to improve network quality. Ref. [72] obtained a more efficient data exchange process by adjusting the size of CWs for vehicular ad hoc networks. The proposal could extract channel characteristics and predict the transmission success rate of packets. The simulations showed that the proposed scheme performed better than the existing backoff schemes. Ref. [73] tried to use the RL model to adjust the CW size by counting the current network throughput, while [74] tracked the wireless traffic load to promote the RL model for a novel CSMA/CA-based MAC. In [75], the authors proposed a novel RL model based on channel observation to predict the future channel state by observing whether the channel state was busy or idle and reduced the collision of access channels in dense WNs by controlling the size of the CW. A wireless smart utility network (Wi-SUN) is a large-scale smart city network that uses unslotted CSMA/CA. With the increase of the network scale, the competition becomes more serious, and the performance of a wireless MAC protocol will deteriorate rapidly. Ref. [76] proposed a *Q*-learning-based unslotted CSMA/CA (QUC) scheme to select the best backoff delay. The performance of the packet delivery ratio was improved by 20%.

Due to the poor performance of the latest CSMA scheme in multihop scenarios, Ref. [37] proposed a scheme to adopt an experience-driven approach and train a CSMA-based MAC with a DRL algorithm named Neuro-DCF. The solution adopted a multiagent RL framework for stable training with a distributed execution and introduced a new graph neural network (GNN)-based training construction for training a unified approach that incorporated various perturbation patterns and configurations. A GNN is a network that is used to process graph-type data. The representation of each node in a graph is calculated from the characteristics of the node, the characteristics of the edges connected to the node, the neighbor representation of the node and the characteristics of its neighbor nodes. That scheme comprehensively investigated multiple factors in the network, such as throughput, delay and channel utilization, which can accelerate the speed of decision convergence. Simulation showed that the scheme could improve the delay performance while preserving optimal utility.

Besides the channel allocation schemes, energy consumption is also considered as a solution for network performance improvement. Ref. [77] proposed an RL-based CSMA scheme for energy saving, which took into account the input and output of a power control method and multiple variables at the PHY layer. A similar study was proposed in [78], which introduced a deep-Q-learning-based scheme named ambient energy harvesting CSMA/CA (AEH-CSMA/CA) for the IoT. This scheme obtained the energy of the environment to intelligently adjust the size of its initial backoff window during the backoff process. The simulation results showed that the throughput of this scheme was 20% higher than IEEE 802.11 with a low energy supply. The above-discussed MAC protocols are summarized in Table 3.

#### 4.3.2. Scheduling

In cognitive ad hoc networks lacking a central controller, the secondary users (SUs) and the primary users (PUs) interfere with each other and synchronization is difficult. To overcome the above problems, Ref. [33] proposed a multichannel cognitive MAC protocol based on distributed RL for opportunistic spectrum access scheduling. This learning mechanism selected channels based on SUs’ detection of PU traffic, avoided collisions with the PUs and kept SUs synchronized. Due to the burstiness of PUs’ traffic, SUs needed to detect transmission opportunities in channels, and each SU employed an LA mechanism whose actions were updated based on environmental feedback from each channel. Having the same PU between clusters of nodes consisting of SUs, they could exchange their PU’s existence experience within the cluster. Meanwhile, the protocol employed a control channel for resource reservation to handle interference and hidden terminal issues between SUs. Compared with the transmission opportunity through the statistical mechanism, the channel utilization of the proposed protocol was improved by 40% and the collision with the PUs was reduced.

In [43], the author proposed a novel *p*-persistent CSMA protocol based on RL by optimizing channel access scheduling. The channel allocation for SUs is a challenging issue due to packet collisions, while SUs may degrade the performance of PUs. This scheme achieved better channel utilization with reduced collisions for SUs by decreasing *p* in the presence of collisions and increasing *p* for successful transmissions. Meanwhile, this method maintained the throughput of the PUs and improved the channel utilization rate through sharing the PUs and SUs’ traffic. The results showed that when the PUs increased the delay, the SUs could effectively use the available channels. Considering the transmission probability greatly affects the network throughput in WNs using *p*-persistent CSMA schemes, Ref. [79] proposed a multistate RL scheme. This scheme learned its optimal strategy by sensing the channel history information, especially the number of collisions or successful transmissions. Simulation showed that the average transmission success rate of this scheme was 45% higher than the traditional schemes.

The performance of the IEEE 802.15.4 MAC depends on the correct configuration of MAC parameters. In [80], an optimization method for adaptively configuring IEEE 802.15.4 MAC parameters was proposed. This scheme determined the MAC parameters based on the channel traffic status and channel characteristics with the predictive feature of RL algorithms and hence supported the dynamic nature of communications. The proposed solution built a learning model that could obtain optimized parameters, including sending rate frequency and packet interarrival time by considering a multilayer perceptron and random forests. Compared to the IEEE 802.15.4 standard with default parameters, this scheme reduced the end-to-end delay and ensured a stable throughput.

The performance of RL schemes relying on timely feedback rewards is severely limited because of the poor delay in UANs. An improved two-stage *Q*-learning scheme was proposed to obtain hidden reward for UANs, named the packet flow ALOHA with *Q*-learning (ALOHA-QUPAF) MAC protocol [11]. The two stages in that scheme were denoted as slot selection and flow harmony. They could not transmit and receive simultaneously in a UAN with half-duplex communication. Therefore, this scheme penalized these received slots to avoid collisions by reducing their *Q*-values. In a healthy channel, there is a continuous flow of packets on the chain. Thus, when the first packet is received, the receiver expects a packet in every frame that follows. As long as the packet flow is interrupted, a packet collision is inferred. The scheme isolated the receiving slot from the transmitting slot to avoid collisions by modifying the scheduling in the MM.

An RL-based MAC protocol for multimedia sensing in UANs was proposed in [12]. It improved the efficiency of mobile relay nodes by using transmission opportunity (TXOP) for relay nodes in multihop UANs. Transceiver nodes and relay nodes were allocated traffic demands based on the competition of sensors using the initial phase of *Q*-learning. Moreover, this solution allocated TXOP resources for the uplink devices based on the traffic demands. The simulations showed that the scheme had an efficient packet delivery rate and throughput.

To control the data transmission rate of nodes in CSMA/CA systems, Ref. [81] designed an RL scheme. By learning the timeout events of packets in the environment, the agent selected actions to control the data transmission rate of the node and adjusted the modulation and coding scheme (MCS) level of data packets to effectively use the available bandwidth in the dynamic channel conditions. Similarly, Ref. [82] also proposed a *Q*-learning system to control the data transmission rates for CSMA/CA-based systems. The RL agent tried to collect channel information and control the MCS level of data packets to obtain better network throughput.

When nodes become active and inactive in the WSN as nodes randomly join and leave the network, the long-term average network age of information (AoI) of their respective processes can be collectively minimized at the remote monitors. An optimized modification of the ALOHA-QT algorithm was proposed in [35], which employed a policy tree (PT) and RL to achieve high throughput. A PT is a prediction model, which represents a mapping relationship between attributes and values. The proposed algorithm was provided with an upper bound on the average AoI and a pointer to select its key parameters. The simulations showed that the proposed algorithm outperformed ALOHA-QT in terms of AoI while requiring less consumption and computation. The above schemes based on scheduling changes are summarized in the Table 4.

#### 4.3.3. Other MAC Mechanisms

A smart-CSMA (SCSMA) MAC scheme with control messages was proposed for low-power and low-bandwidth WSNs in [44]. The 802.15.4 MAC does not use control messages for transmission due to the small size of the data, and the cost of control messages is relatively high. The proposal assigned different start delay slot numbers (SDNs) to neighboring nodes by making each node rebroadcast a message. A node was designed to collect the above data to reduce collisions from neighbor nodes and hidden terminals. For two competing hidden users, their waiting time difference was adjusted to be greater than the sum of the transmission times of RTS and CTS. Therefore, the busy channel status was detected by users with a longer waiting time. The blind learning (BL) algorithm was proposed to optimize the exchange of messages. By listening the control messages, BL-based agents adjusted their SDN intelligently with hidden terminals or topology changes. The result showed that this scheme appropriately adjusted its SDN to improve the throughput by 42.1% and reduced the energy consumption by 15.7%.

In [39], a distributed-RL MAC protocol with localized information in WSNs was proposed. In order to maximize the network throughput, the solution adopted distributed learning of the optimal MAC transmission strategy among nodes. Moreover, nodes also obtained channel information from the closest two-hop neighbor nodes and adaptively adjusted their transmission strategy to maintain maximum throughput under dynamic topology loads. This solution was also designed to parametrically change the access priority of nodes while learning, which is very practical for WANET applications with different priorities. The results showed that the performance of this scheme was improved by 20%.

Many RL solutions proposed so far assume real-time feedback of the results of packet transmission. In [83], a novel RL-based MAC protocol ALOHA-dQT was proposed, which improved the channel utilization by letting nodes periodically broadcast short summaries of their known channel history. The channel history for the last *N* slot states was stored by each node, and it iteratively merged its information with the channel history based on the node broadcast. The RL system updated the relevant information to make the node adaptive.

Ref. [23] proposed a novel RL-based scheme for FANET, named position-prediction-based directional MAC (PPMAC). This scheme addressed the directional deafness problem by using position information, which is very practical for FANET applications in a dynamically changing environment. The results showed that the reward of this learning system was improved by 16%.

In addition, in order to achieve a better end-to-end QoS and ensure fairness in user-dense scenarios, work on the optimization of learning algorithms has also been proposed. In [84], a novel Monte Carlo (MC) reward update scheme for DRL training was proposed. This scheme used the access history of each site to derive the DRL-based MAC protocol. Compared with the traditional distributed coordination function (DCF), this scheme could effectively improve the network throughput. The above-discussed MAC protocols are summarized in Table 5.

### 4.4. Summary

We can find that most of the proposals try to improve the MMs by obtaining medium access opportunity and then performing optimal scheduling. The traditional medium access schemes cannot well meet the rapid changes of the wireless network topologies, while RL solutions can learn what the regular nodes’ behavior is by creating models from past scenarios. Modifying the MAU has been proposed, such as assigning subchannels to make networks have a better performance, and with scheduling, the sender makes decisions locally to reduce collision from different senders.

Meanwhile, RL schemes are often used to optimize the channels, as some of the channel variations occur in a changeable environment, to let the node learn and make real-time decisions tailored to each situation, especially with the advantage of enabling nodes to perceive and interact with the environment.

## 5. Open Research Issues

Here, we discuss some challenging issues following the MAC reference model and give some directions for future research in this section. A summary of the discussed is presented in Table 6.

### 5.1. Operation Cycle

Both large frame and superframe divide the time into several segments. The former allocates each segment equally to the nodes and uses the same access scheme, while the latter adopts different access schemes for each segment, which is mainly used in scenarios that need to run multiple transmission methods at the same time, such as prioritization and emergency transmission. In this regard, RL solutions can be used for adaptive selective access schemes. In addition, coordinated scheduling with neighbors can use RL schemes to adapt the changing networks and topologies in future research, e.g., for nodes to learn the parameters selection and the extent of dynamic network changes.

### 5.2. Medium Access Unit

Several schemes have been proposed in certain functions, such as adjusting the slot length to handle the channel access and channel allocation with multiple access schemes. However, there are still some issues to be discussed. For example, with TDM, time is divided into regular time units. Although collisions are avoided by inserting a guard time between two adjacent time slots, it reduces bandwidth utilization. RL schemes can be used by nodes to adaptively adjust the size of the guard time according to the distribution of active nodes. With learning individual node patterns and requirements, RL schemes can help agents to learn which slot should be accessed to maximize bandwidth utilization. Another issue in the MAU is data channel allocation in multichannel protocols. Since a node can only work either on the control channel or the data channel, but not both at the same time, RL schemes can be applied to learn individual node’s channel conditions and requirements to allocate channels required by the node without the control channel.

### 5.3. MAC Mechanism

A MM can be used alone to prevent collisions, or multiple MMs can be combined to form a new complicated mechanism [55]. State-of-the-art RL-based works focus on tuning the MMs’ parameters. An issue is how to choose those MMs reasonably to effectively improve the MAC protocol performance. RL schemes can be used to choose MMs and form them to design a MAC protocol for different network scenarios by creating RL models and learning from current and past network status.

### 5.4. Other Issues

In WANETs, there are usually no nodes dedicated to coordinate communication, and each node dynamically forms a temporary network due to mobility. Mobility is one of the main factors affecting network performance since it produces a frequently changing network topology. In the following topics, some applications of RL for adaptability are discussed.

(1) Environmental prediction: The wireless network environment is complex and changeable; RL schemes are used to discover the rules of potential changes between environmental characteristics and reduce the impact of uncertain factors on the protocol performance to make better real-time decisions. For example, a method called spatial reuse, which enables two adjacent transmitters to send signals simultaneously without affecting each other. It is feasible to exploit similar features to maximize channel utilization as much as possible by using RL.

(2) Resource allocation and task scheduling: Channel resources are usually limited, and actual power allocation and channel constraints are the main causes of message collisions. The latest mathematical optimization theory still lacks a systematic resource allocation method, and resource scheduling will increase the complexity of the protocol. The protocol design can further use RL schemes to improve the capabilities of nodes, by decreasing complexity and high overheads.

(3) Systematical cross-layer design: Some MAC protocols using a cross-layer design also focus solely on MAC performance and ignore the performance degradation of the network as a whole. For example, the physical layer data is employed to detect channel variation and quality; meanwhile, the network layer data, for instance congestion, is employed to determine the status of an entire link. Using the powerful data aggregation function of RL, a protocol design can comprehensively consider all available optimization options to enhance performance [55].

(4) Underwater acoustic networks: Many modifications have noticed the drawbacks of long propagation delays in UANs and tried to turn the drawbacks of long delays into advantages by setting relevant parameters [55]. However, some protocols do not thoroughly consider the peculiar features of UANs, such as an unstable channel, rapidly changing channel states and the leveraging of long propagation delay. With RL, these features can be fully considered as much as possible to reduce the probability of signal collision.

## 6. Conclusions

This paper reviewed the methodologies of applying an RL scheme in WNs within the context of the MAC reference model. Although some works have been conducted in developing RL in MAC protocol design, there are still some open issues that urgently need to be addressed in this field, covering some directions according to different MAC components, especially the MMs. We hope that this article will help the reader to understand the novel RL-based MAC schemes and provide a guide for researchers to optimize WNs using RL techniques.

## Figures and Tables

**Figure 1 entropy-25-00101-f001:**
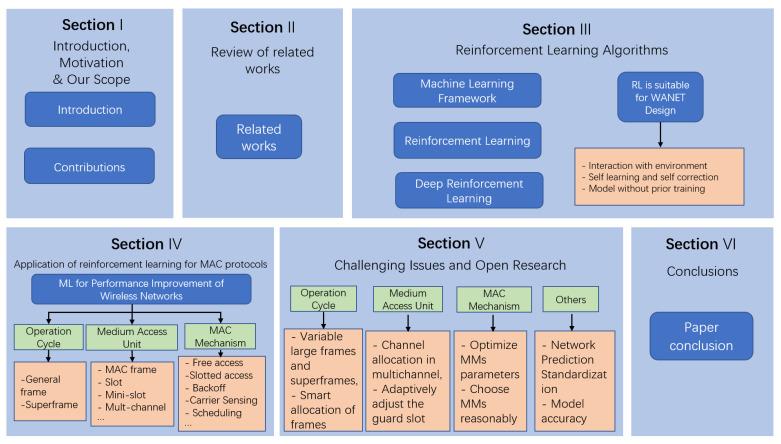
Paper outline.

**Figure 2 entropy-25-00101-f002:**
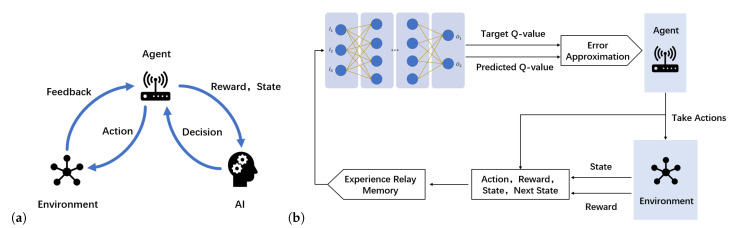
Schematic diagrams of RL and DRL. (**a**) Schematic diagram of reinforcement learning. (**b**) Schematic diagram of deep reinforcement learning [54].

**Figure 3 entropy-25-00101-f003:**
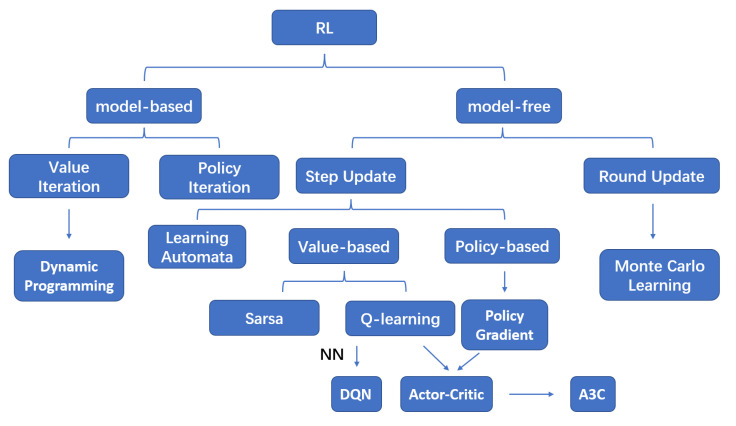
A taxonomy of reinforcement learning.

**Figure 4 entropy-25-00101-f004:**
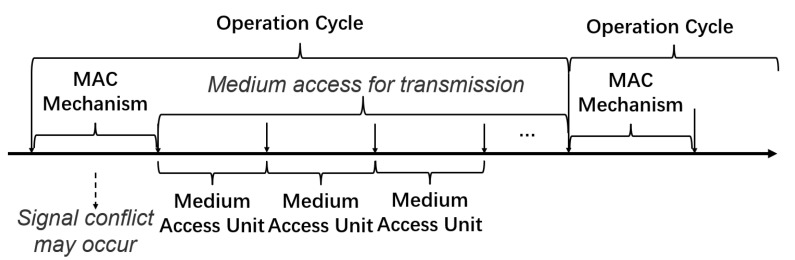
The architecture diagram of MAC protocol components [55].

**Table 1 entropy-25-00101-t001:** Comparison between our survey and the related existing ones.

References	Networking Schemes	Learning Model				Survey Contents
		RL	DL	DRL	Other	
[56] (2019)	Cellular, ad hoc, cognitive radio networks (CRNs), IoT		✓	✓		The crossovers between DL models and mobile/wireless networking
[57] (2017)	WSNs, IoT	✓			✓	ML algorithms for WSNs.
[58] (2022)	TDMA, IoT	✓			✓	RL-based schemes for TDMA-based MAC protocols.
[59] (2021)	IoT, mobile big data, CRNs, WSNs	✓		✓	✓	RL-based schemes for WNs of three layers: PHY, MAC and network.
[31] (2022)	1G to 6G				✓	Multiple access schemes for next-generation wireless.
[32] (2021)	WNs	✓	✓		✓	ML-based methods for ensuring end-to-end QoS and QoE.
[60] (2019)	Cellular, WNs	✓			✓	ML-based schemes for network access, routing for traffic control and streaming adaption in WSNs.
[61] (2021)	RIS-aided WNs				✓	AI-assisted MAC for RIS-aided WNs.
[62] (2021)	WSN, IoT	✓			✓	RL-based methods in WSNs for smart city applications.
[63] (2019)	UAV, WNs				✓	ML framework for UAV-based communications.
This paper	WSNs, IoT, cellular, ad hoc, CRNs, USNs, HetNet, UANs	✓	✓	✓		RL/DRL-based MAC protocols for WNs with a MAC reference model.

**Table 2 entropy-25-00101-t002:** Comparison of the RL techniques used in OC and MAU.

Protocols (References)	MAC Components	Network	Algorithm	Data Used for Learning	Performance
eOQ-MAC [46] (2019)	Superframe (OC)	WNs	*Q*-learning	Packet loss rate, emergency time slot, throughput	Lowered packet loss rate for emergency data transmission.
UA-Slotted CSMA [10] (2013)	Slot (MAU)	UA-WSNs	*Q*-learning	Lifetime, channel status, energy consumption, bandwidth	Minimized power consumption and extended the life of the network.
RL-IRSA [45] (2020)	MAC frame, slot (MAU)	CRNs	*Q*-learning	Convergence time, channel status, throughput	Significantly reduced convergence time with optimized degree distributions for small frame sizes.
DLMA [28] (2019)	Slot (MAU)	HetNet	DRL	Convergence time, throughput	Maximized the total throughput with faster learning speed.
Q-CSMA [71] (2021)	Slot (MAU), scheduling (MM)	WNs	*Q*-learning	Channel status, packet loss rate, delay	Reduced the number of collisions and packet delay.

**Table 3 entropy-25-00101-t003:** Comparison of the RL techniques used for the backoff mechanism.

Protocols (References)	MAC Components	Network	Algorithm	Data Used for Learning	Performance
Distributed rate adaptive CSMA/CA [8] (2016)	Backoff (MM)	WNs	*Q*-learning	Channel status, CW, energy consumption	Enabled users to reduce energy consumption based on their latency-limited needs with faster convergence.
Backoff Improvement [72] (2020)	Backoff (MM)	Vehicular networks	*Q*-learning	Channel status, transmission success rate	Had a more efficient data exchange process and ensured fairness.
CW adjustment scheme [73] (2021)	Backoff (MM)	WNs	*Q*-learning, supervised learning	CW, throughput	Effectively reduced the collision and improved the system throughput.
Performance enhancement CSMA [74] (2021)	Backoff, Scheduling (MM)	WNs	Reinforcement learning	Channel status, energy consumption, traffic load	Had a stable throughput.
Channel-observation-based MAC [75] (2018)	Backoff, scheduling (MM)	Dense WLANs	DRL	CW, throughput, channel status	Efficiently predicted the future channel state.
QUC [76] (2022)	Backoff (MM)	Wireless smart utility networks	*Q*-learning	Throughput, delay	The performance of the MAC layer was improved by 20%.
Neuro-DCF [37] (2021)	Backoff (MM), Slot(MAU)	WNs	DRL + GNN	Throughput, delay, channel utilization	Reduced the end-to-end delay while preserving optimal utility.
AEH-CSMA/CA [78] (2020)	Backoff (MM) (2020)	IoTs	Deep *Q*-learning	Throughput, CW, energy	Ensured a high throughput and low energy supply.

**Table 4 entropy-25-00101-t004:** Comparison of the RL techniques used for the scheduling mechanism.

Protocols (References)	MAC Components	Network	Algorithm	Data Used for Learning	Performance
MCC-MAC [33] (2015)	Scheduling (MM)	CRNs	*Q*-learning + LA	Network traffic, channel status	Avoided the conflict between SUs and PUs.
*p*-persistent CSMA [43] (2011)	Scheduling (MM)	CRNs	*Q*-learning	Throughput, channel utilization	Had good robustness and the SU efficiently utilized the available channel at the expense of an extra delay of the PU.
Optimal parameters for IEEE 802.15.4 [80] (2020)	Scheduling (MM)	WNs	Supervised learning	MAC parameters, delay, channel status	Increased the dynamic adaptability of nodes.
ALOHA-QUPAF [11] (2021)	Scheduling (MM)	UA-WSNs	Modified *Q*-learning	Throughput, channel status	Isolated the receiving slot from the transmitting slot to avoid collisions.
RL-MAC [12] (2022)	Scheduling (MM)	UA-WSNs	*Q*-learning	Channel traffic, energy consumption	Improved throughput with limited energy.
Improved ALOHA-QT [35] (2022)	Scheduling (MM)	WNs	RL + PT	Throughput, AoI, energy consumption	Adapted to the changing number of active agents with less energy.
Rate adaptation scheme [81] (2020)	Scheduling (MM)	WNs	*Q*-learning	MCS channel utilization, bandwidth	Obtained better network throughput.
Improved *p*-persistent CSMA [79] (2018)	Scheduling (MM)	WNs	Multi-state *Q*-learning	Channel status	Investigated the application of multistate RL algorithm.

**Table 5 entropy-25-00101-t005:** Comparison of the RL techniques used for other mechanisms.

Protocols (References)	MAC Components	Network	Algorithm	Data Used for Learning	Performance
SCSMA [44] (2020)	Messaging (MM)	WSNs	Blind learning [44]	SDN, throughput, channel status	Improved throughput, reduced energy consumption and could avoid hidden terminal problems.
Distributed ALOHA [39] (2022)	Prioritization (MM)	IoTs	*Q*-learning	Throughput, channel status	Maintained maximum throughput under dynamic topology loads.
ALOHA-dQT [83] (2020)	Messaging (MM)	WNs	Reinforcement learning	Channel history, throughput	Achieved a high channel utilization.
PPMAC [23] (2018)	Messaging (MM)	FANET	*Q*-learning	Channel status, position information	Provided an intelligent and highly adaptive communication solution.
SPCA [85] (2019)	Messaging (MM)	WSNs	DRL	Spectrograms of TDMA, channel utilization	Reduced the number of collisions with efficient channel utilization.

**Table 6 entropy-25-00101-t006:** A summary of open research issues.

MAC Components	Open Research Issues
Operation cycle	Variable large frames and superframes Smart frame allocation algorithms
Medium access unit	Channel allocation in multichannel Adaptively adjust the guard slot
MAC mechanism	Optimize MMs’ parameters Choose MMs reasonably
Other issues	Network prediction standardization and model accuracy Resource allocation and task scheduling Systematical cross-layer design Fully consider the features of UANs

## Data Availability

Not applicable.

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
