# Peer review of "Survey of Reinforcement-Learning-Based MAC Protocols for Wireless Ad Hoc Networks with a MAC Reference Model"

_entropy, 2023, doi:10.3390/e25010101_

Round 1
Reviewer 1 Report (Previous Reviewer 3)
the idea is good but the paper should be improved.
1) Please add some diagrams about the references. For example the diversity of each method or the years of publication.
2) More information about each method may give some valuable information. Adding some classification or taxonomy may be usefull. The author may extract some of these items from table2.
3) Extracting some open research may also be considered in this paper.
4) If there is connection between summarized methods with classical problems such as graph coloring please add them.
5) The refrences to LA based works were not accurate! i hope the rest of them should be ok!
Author Response
Dear Reviewer,
We would like to thank you for your efforts in reviewing our manuscript titled "Survey of Reinforcement Learning Based MAC Protocols for Wireless Ad Hoc Networks with a MAC Reference Model", and providing many helpful comments and suggestions to improve our paper and to guide our research in the future. We have studied your comments point by point, and revised the manuscript accordingly as reported below. All authors have approved the response letter and the revised version of the manuscript.
- Please add some diagrams about the references. For example the diversity of each method or the years of publication.
Response: The tables were modified according to the comment. We add a column to the tables to introduce the data used by the learning system, and the years of publication have been added. (Table.2 ,3,4,5)
- More information about each method may give some valuable information. Adding some classification or taxonomy may be usefull. The author may extract some of these items from table2.
Response: We present Section 4.3 with subsections for different MMs and give the corresponding table (Table.2 ,3,4,5). We give the classification of RL in Fig .3 page 6, and add relevant description in Section 3 (Line132-138, 150-151166-172, Page 4-6).
- Extracting some open research may also be considered in this paper.
Response: We present a table to summarize the discussed open research issues. (Page16, Table. 6)
- If there is connection between summarized methods with classical problems such as graph coloring please add them.
Response: The wireless network environment model can be regarded as Markov process, Therefore, we briefly introduce MDP. (Line 166, Page 5). In addition, GNN and PT related to references were also mentioned. (Line 320-325 Page 9, Line 405-406,Page 13)
- The refrences to LA based works were not accurate! i hope the rest of them should be ok!
Response: We fix it and add the correct references. (Line 166, Page 5)
We would like to take this opportunity to thank you for all your time involved and this great opportunity for us to improve the manuscript. We hope you will find this revised version satisfactory.
Yours sincerely,
Zhichao Zheng
Reviewer 2 Report (New Reviewer)
This paper presents a survey of the literature about reinforcement learning (RL) based medium access control (MAC) protocols for wireless ad hoc network (WANET). I have the following concerns:
1. According to table 1, the novelty is low. Novelty should be increased by considering more networks and learning models.
2. You can add different mobile ad hoc networks such as vehicular ad hoc networks [i], Flying ad-hoc networks [ii], etc. You can also mention about next-generation wireless communication [iii, iv].
3. RL and DRL learning models are considered. DL can be added.
4. In Section 4, different performance metrics such as throughput, delay, channel utilization, etc. should be discussed with mathematical representation. In Table 2, it is mentioned which paper considers which performance metrics.
5. Section 4.3 and Table 2 are long. It would be nice to present Section 4.3 with subsections for different ad hoc networks and give the corresponding table there.
6. A table can be given for Section 5 too.
7. It is difficult to follow. Please upload a fresh copy of the manuscript. The modified text or paragraph only can be highlighted.
Author Response
Dear Reviewer,
We would like to thank you for your efforts in reviewing our manuscript titled "Survey of Reinforcement Learning Based MAC Protocols for Wireless Ad Hoc Networks with a MAC Reference Model", and providing many helpful comments and suggestions to improve our paper and to guide our research in the future. We have studied your comments point by point revised the manuscript accordingly as reported below. All authors have approved the response letter and the revised version of the manuscript.
- According to table 1, the novelty is low. Novelty should be increased by considering more networks and learning models.
Response: We have added some reviewed networks, such as vehicular ad hoc networks, FANET and next-generation wireless communication.(Lines 43-44,92-93, page 2-3) We also give the classification of RL in Fig .3 page 6, and add relevant description in Section 3 (Lines 132-138, 150-151166-172, Page 4-6).
- You can add different mobile ad hoc networks such as vehicular ad hoc networks [i], Flying ad-hoc networks [ii], etc. You can also mention about next-generation wireless communication [iii, iv].
Response: We add some relevant references in Line 43-44,92-93, page 2-3.
- RL and DRL learning models are considered. DL can be added.
Response: The latest integration technology of DL and RL is very common. We have introduced DL and this integration process in Section 3.3 (Lines 166-172, Page 5).
- In Section 4, different performance metrics such as throughput, delay, channel utilization, etc. should be discussed with mathematical representation. In Table 2, it is mentioned which paper considers which performance metrics.
Response: We apply your advice to express our review more detail. Performance metrics have been discussed with mathematical representation. (Lines 239-240,297,314,335, etc.) We mentioned that the performance metrics considered in the references are in the column Performance. In addition, we add a column to the tables to introduce the data used by the learning system, and the years of publication have been added. (Table.2,3,4,5)
- Section 4.3 and Table 2 are long. It would be nice to present Section 4.3 with subsections for different ad hoc networks and give the corresponding table there.
Response: The review here adopts the MAC reference model proposed in our early work, which compartmentalizes a MAC protocol into the follows three components: operation cycle (OC), medium access unit (MAU) and MAC mechanism (MM). Following this model, how RL schemes are used to improve the performance and effectiveness of there components are discussed. Therefore, we present Section 4.3 with subsections for different MMs and give the corresponding table. (Table.2,3,4,5)
- A table can be given for Section 5 too.
Response: This table was added according to the comment. (Page16, Table. 6)
- It is difficult to follow. Please upload a fresh copy of the manuscript. The modified text or paragraph only can be highlighted.
Response: We modified the automatically generated diff file to make it easier to follow.
We would like to take this opportunity to thank you for all your time involved and this great opportunity for us to improve the manuscript. We hope you will find this revised version satisfactory.
Yours sincerely,
Zhichao Zheng
Round 2
Reviewer 1 Report (Previous Reviewer 3)
the response note has not matched the paper. the paper has no table 6!! figure 3 is not a classification. please re-upload the file!
Reviewer 2 Report (New Reviewer)
I think the authors have addressed my comments well. I have no further comments.
This manuscript is a resubmission of an earlier submission. The following is a list of the peer review reports and author responses from that submission.
Round 1
Reviewer 1 Report
The paper does not present any specific or quantitative estimates such as:
How do these implimentation scale?
Scaling limits?
Quantitative examples and comparisons to exisiting legacy implementations would be advantageous.
Author Response
Dear editor reviewer,
We would like to thank you for your efforts in reviewing our manuscript titled "Survey of Reinforcement Learning Based MAC Protocols for Wireless Ad Hoc Networks with a MAC Reference Model", and providing many helpful comments and suggestions, which will all prove invaluable in the revision and improvement of our paper, as well as in guiding our research in the future. We have studied your comments point by point, revised the manuscript accordingly. The amendments are highlighted in blue in the revised manuscript. All authors have approved the response letter and the revised version of the manuscript.
We really appreciate you for your carefulness and conscientiousness. Your suggestions are really valuable and helpful for revising and improving our paper. According to your suggestions, we have made the following revisions on this manuscript:
1.The paper does not present any specific or quantitative estimates.
Response:We are grateful for the suggestion. We added the section.3 in line175-215 page7 showing existing legacy implementations and explaining why RL can better advance MAC design.
Thank you again for your valuable comments and suggestions. I look forward to hearing from you soon in due course.
Yours sincerely,
Zhichao Zheng
Reviewer 2 Report
This study discusses about the limitations by addressing the issues of traditional MAC protocol for Wireless Adhoc network. Overall, the work is fine, however, there are some issues that must be resolve before the publication.
1. The plagiarism is too high. Cannot be acceptable in this form.
. . The motivation in the Abstract is missing.
2. The introduction part is week, add some more study about Adhoc network and MAC protocols.
3. The author should highlight the current studies which have done conduction on similar area. They should discuss the limitation of current and then discuss their motivation of this work,
4. There should be one figure at the end of Chapter which show the Chapter organization.
5. There should be some studies related to Blind Learning Scheme.
6. Paper organization paragraph is missing. The overall paper organization must be given in a separate paragraph (last para) of introduction section.
7. Check all the Abbreviations.
8. The figures size is very big. Revise all figures with appropriate visibility.
9. Should add some latest reference of 2021 and 2022.
1. Author should proofread and edit their manuscript for clarity.
Author Response
Dear reviewer,
We would like to thank you for your efforts in reviewing our manuscript titled "Survey of Reinforcement Learning Based MAC Protocols for Wireless Ad Hoc Networks with a MAC Reference Model", and providing many helpful comments and suggestions, which will all prove invaluable in the revision and improvement of our paper, as well as in guiding our research in the future. We have studied your comments point by point, revised the manuscript accordingly. The amendments are highlighted in blue in the revised manuscript. All authors have approved the response letter and the revised version of the manuscript.
We really appreciate you for your carefulness and conscientiousness. Your suggestions are really valuable and helpful for revising and improving our paper. According to your suggestions, we have made the following revisions on this manuscript:
- The motivation in the Abstract is missing.
Response:Thank you very much for your advice. We have added the information required as explained above (Lines 2-8, page 1).
- The introduction part is week, add some more study about Adhoc network and MAC protocols.
Response:Thank you for your constructive comments. We have added some more latest reference about Adhoc network and MAC protocols above (Lines 49-51, page 2) (we can not highlight the reference due to limitations of the "Track Changes" function. ). In Lines 52-55, page 2 , this phrase was modified according to the comment. Moreover, we have added the section.3 in line175-215 page7 to introduce more study about Adhoc network.
- The author should highlight the current studies which have done conduction on similar area. They should discuss the limitation of current and then discuss their motivation of this work
Response:Thank you for your positive comments. We added the section.3 in line144-215 page 6-7 showing existing legacy implementations and explaining why RL can better advance MAC design.
- There should be one figure at the end of Chapter which show the Chapter organization.
Response:Thank you for your insightful comment. We have added the figure in Fig.1 page3 required as explained above.
- There should be some studies related to Blind Learning Scheme.
Response:Thank you very much for your valuable comments. We have added the phrase and the algorithm in line335-338 page 10 to introduce Blind Learning Scheme.
- Paper organization paragraph is missing. The overall paper organization must be given in a separate paragraph (last para) of introduction section.
Response:Thank you for underlining this deficiency. We have added the paper organization required as explained above (Lines 70-77, page 2).
- Check all the Abbreviations.
Response:Thank you for your positive comments. We have fixed in line489-497 page 14.
- The figures size is very big. Revise all figures with appropriate visibility.
Response:Thank you for your insightful comment. The figures (Fig.2-5)were revised and modified according to the comment
- Should add some latest reference of 2021 and 2022.
Response:We are grateful for the suggestion. We have added some latest reference in Lines 360-404, page 12-13 required as explained above. The Table.1 has been revised based on comments in page 13.
- Author should proofread and edit their manuscript for clarity.
Response:Thank you very much for your advice. Modified throughout the text according to the comment, All corrected spellings have been highlighted in blue.
Thank you again for your valuable comments and suggestions. I look forward to hearing from you soon in due course.
Yours sincerely,
Zhichao Zheng
Reviewer 3 Report
The title is very interesting but the following problem should be considered.
The paper has low cohesion considering its title. sometimes it tries to cover reinforcement learning and sometimes it tries to cover other learning algorithms
The abstract is a general statement and it should be reorganized to focus on the results of the survey.
Figure 1 is vague. is it a representative example of a wireless network? which elements of these networks are covered by that figure.
Machine learning algorithms are not limited to neural networks and reinforcement learning.
the descriptions of reinforcement learning and neural network in comparison with wireless parts are not fair. more information about the challenges of wireless networks should be given.
From a learning perspective, reinforcement learning contains a wide range of algorithms and models such as learning automata. by removing the subclasses of the learning algorithm the survey will be weak. please consider this issue during revision.
The number of references for this paper as a survey paper is not appropriate.
I expected to see some taxonomies and classifications as a result of this paper. I don't think that table 1 covers all related works
Author Response
Dear reviewer,
We would like to thank you for your efforts in reviewing our manuscript titled "Survey of Reinforcement Learning Based MAC Protocols for Wireless Ad Hoc Networks with a MAC Reference Model", and providing many helpful comments and suggestions, which will all prove invaluable in the revision and improvement of our paper, as well as in guiding our research in the future. We have studied your comments point by point, revised the manuscript accordingly. The amendments are highlighted in blue in the revised manuscript. All authors have approved the response letter and the revised version of the manuscript.
We really appreciate you for your carefulness and conscientiousness. Your suggestions are really valuable and helpful for revising and improving our paper. According to your suggestions, we have made the following revisions on this manuscript:
- The paper has low cohesion considering its title. sometimes it tries to cover reinforcement learning and sometimes it tries to cover other learning algorithms
Response:Thank you for your constructive comments. In fact,we tries to cover reinforcement learning. This sentence was rephrased according to the comment (Line 56-63, page 2).
- The abstract is a general statement and it should be reorganized to focus on the results of the survey.
Response:Thank you very much for your advice. This section was revised and modified according to the information showed in the work suggested by the reviewer (Line 36-37,52-58,70-77, page 2).
- Figure 1 is vague. is it a representative example of a wireless network? which elements of these networks are covered by that figure.
Response:Thank you for your positive comments. we removed it.
- Machine learning algorithms are not limited to neural networks and reinforcement learning.
Response:Thank you for your insightful comment. we focus on RL and DL algorithms. Moreover, we added the section.3 in line189-215 page7 explaining why RL can better advance MAC design.
- the descriptions of reinforcement learning and neural network in comparison with wireless parts are not fair. more information about the challenges of wireless networks should be given.
Response:Thank you very much for your valuable comments. line175-215 page7 showing existing legacy implementations in WANETs.
- From a learning perspective, reinforcement learning contains a wide range of algorithms and models such as learning automata. by removing the subclasses of the learning algorithm the survey will be weak. please consider this issue during revision.
Response:Thank you for the suggestion. We have added the phrase and the algorithm in line335-338 page 10 to introduce Blind Learning Scheme.
- The number of references for this paper as a survey paper is not appropriate.
Response:We are grateful for the suggestion. We have added some latest reference in Lines 360-404, page 12-13 required as explained above.
- I expected to see some taxonomies and classifications as a result of this paper. I don't think that table 1 covers all related works
Response:Thank you for your insightful comment. we classify the reviewed RL-based MAC protocols according to a MAC reference model, and discuss how each proposed solution works, and the challenges issues on the related MAC model components. The Table.1 has been revised based on comments in page 13.
Thank you again for your valuable comments and suggestions. I look forward to hearing from you soon in due course.
Yours sincerely,
Zhichao Zheng
Round 2
Reviewer 2 Report
The authors IGNORE THE PLAGAIRISM comments. It still high +40%. CANNOT BE ACCEPTED LIKE THIS. MUST REDUCE TO 15% and less.
Other than that, the reviewer is satisfied with the changes provided by authors. However, the authors can cite some recent related papers such as:
1) Mobility and queue length aware routing approach for network stability and load balancing in MANET (2021)
2) Performance analysis of ad hoc on-demand distance vector routing protocol for MANET (2020)
Author Response
Dear reviewer,
Thank you for giving us the opportunity to submit a revised draft of the manuscript "Survey of Reinforcement Learning Based MAC Protocols for Wireless Ad Hoc Networks with a MAC Reference Model", We appreciate the time and effort that you dedicated to providing feedback on our manuscript and are grateful for the insightful comments on and valuable improvements to our paper.
We really appreciate you for your carefulness and conscientiousness. Your suggestions are really valuable and helpful for revising and improving our paper. According to your suggestions, we have made the following revisions on this manuscript:
- The authors IGNORE THE PLAGAIRISM comments. It still high +40%. CANNOT BE ACCEPTED LIKE THIS. MUST REDUCE TO 15% and less.
Response:We are grateful for the suggestion. We have reduced the similarity rate of the manuscript.
- However, the authors can cite some recent related papers
Response:Thank you for your positive comments. We have added two latest reference in Lines 45, 66-67, page 2 required as explained above.
Once again, we thank you for the time you put in reviewing our paper and look forward to meeting your expectations.
Yours sincerely,
Zhichao Zheng
Reviewer 3 Report
The authors answered to most of my questions.
actually, I suggest adding more material to this survey because it can cover many concepts. for example, I know that cognitive networks are in this domain but they are not covered. Learning automata theory is widely used in wireless networks but i couldn't find any relative reference in the text. taxonomies of detailed discussions can be useful to reinforce the article.
Author Response
Dear reviewer,
Thank you for giving us the opportunity to submit a revised draft of the manuscript "Survey of Reinforcement Learning Based MAC Protocols for Wireless Ad Hoc Networks with a MAC Reference Model", We appreciate the time and effort that you dedicated to providing feedback on our manuscript and are grateful for the insightful comments on and valuable improvements to our paper.
We really appreciate you for your carefulness and conscientiousness. Your suggestions are really valuable and helpful for revising and improving our paper. According to your suggestions, we have made the following revisions on this manuscript:
- actually, I suggest adding more material to this survey because it can cover many concepts. for example, I know that cognitive networks are in this domain but they are not covered. Learning automata theory is widely used in wireless networks but i couldn't find any relative reference in the text. taxonomies of detailed discussions can be useful to reinforce the article.
Response:We are grateful for the suggestion. We have added the phrase in line138-145 page 5 to introduce Learning automata scheme. We have added two cognitive networks related references in lines 361-378, page 10-11 and one LA related reference in lines 361-398, page 10-11 required as explained above. Moreover, the Table.1 has been revised based on comments in page 13.
Once again, we thank you for the time you put in reviewing our paper and look forward to meeting your expectations.
Yours sincerely,
Zhichao Zheng
Round 3
Reviewer 3 Report
the concept is very popular and the paper is very short. I put some comments but the final file is still far from a perfect survey.